# Predictors and Outcomes of Healthcare-Associated Infections among Patients with COVID-19 Admitted to Intensive Care Units in Punjab, Pakistan; Findings and Implications

**DOI:** 10.3390/antibiotics11121806

**Published:** 2022-12-13

**Authors:** Zia Ul Mustafa, Sania Tariq, Zobia Iftikhar, Johanna C. Meyer, Muhammad Salman, Tauqeer Hussain Mallhi, Yusra Habib Khan, Brian Godman, R. Andrew Seaton

**Affiliations:** 1Discipline of Clinical Pharmacy, School of Pharmaceutical Sciences, Universiti Sains Malaysia, Gelugor 11800, Pinang, Malaysia; 2Department of Pharmacy Services, District Headquarter (DHQ) Hospital, Pakpattan 57400, Pakistan; 3Department of Medicine, Faisalabad Medical University, Faisalabad 38000, Pakistan; 4Department of Public Health Pharmacy and Management, School of Pharmacy, Sefako Makgatho Health Sciences University, Ga-Rankuwa, Pretoria 0208, South Africa; 5Institute of Pharmacy, Faculty of Pharmaceutical and Allied Health Sciences, Lahore College for Women University, Lahore 54000, Pakistan; 6Department of Clinical Pharmacy, College of Pharmacy, Jouf University, Sakaka 72388, Saudi Arabia; 7Centre of Medical and Bio-Allied Health Sciences Research, Ajman University, Ajman P.O. Box 346, United Arab Emirates; 8Department of Pharmacoepidemiology, Strathclyde Institute of Pharmacy and Biomedical Science (SIPBS), University of Strathclyde, Glasgow G4 0RE, UK; 9Queen Elizabeth University Hospital, Glasgow G51 4TF, UK; 10Scottish Antimicrobial Prescribing Group, Healthcare Improvement Scotland, Glasgow G1 2NP, UK

**Keywords:** COVID-19, intensive care, healthcare-associated infections, Pakistan, mortality, key factors

## Abstract

Healthcare-associated infections (HAIs) have a considerable impact on morbidity, mortality and costs. The COVID-19 pandemic resulted in an appreciable number of hospitalized patients being admitted to intensive care units (ICUs) globally with a greater risk of HAIs. Consequently, there is a need to evaluate predictors and outcomes of HAIs among COVID-19 patients admitted to ICUs. A retrospective study of patients with COVID-19 admitted to ICUs of three tertiary care hospitals in the Punjab province over a five-month period in 2021 was undertaken to ascertain predictors and outcomes of HAIs. Of the 4534 hospitalized COVID-19 patients, 678 were admitted to ICUs, of which 636 patients fulfilled the inclusion criteria. Overall, 67 HAIs were identified among the admitted patients. Ventilator-associated lower respiratory tract infections and catheter-related urinary tract infections were the most frequent HAIs. A significantly higher number of patients who developed HAIs were on anticoagulants (*p* = 0.003), antithrombotic agents (*p* < 0.001), antivirals (*p* < 0.001) and IL-6 inhibiting agents (*p* < 0.001). Secondary infections were significantly higher in patients who were on invasive mechanical ventilation (*p* < 0.001), had central venous access (*p* = 0.023), and urinary catheters (*p* < 0.001). The mortality rate was significantly higher in those with secondary infections (25.8% vs. 1.2%, *p* < 0.001). Our study concluded that COVID-19 patients admitted to ICUs have a high prevalence of HAIs associated with greater mortality. Key factors need to be addressed to reduce HAIs.

## 1. Introduction

Healthcare-associated infections (HAIs) are considered the most frequent adverse consequences of healthcare delivery. They represent a significant public health problem in terms of increasing hospital stay, morbidity and mortality as well as associated costs, including patients with coronavirus disease of 2019 (COVID-19) [1,2,3,4,5,6,7]. The prevalence of HAIs varies considerably across countries [6]. Among high-income countries, in the USA approximately 3 to 4% of patients will develop an HAI, with attributable costs estimated at $3384 ($885–$7717) per patient for *vancomycin-resistant enterococci* up to $39,787 ($20,813–$64,140) for *MDR acinetobacter* and $74,306 ($20,377–$128,235) for *carbapenem-resistant* (*CR*) *acinetobacter* [8,9,10]. Overall, it was estimated in 2016 that healthcare costs associated with HAIs in the US varied between $7.2 billion to $14.9 billion [11]. In China, the additional costs of treating antimicrobial-resistant HAIs were estimated at $15,557.25 per patient compared with non-HAI patients exacerbated by on average an extra 41 days in hospital [12]. 

Lower rates of HAIs were recently reported in Scotland at approximately 1% of acute care patients pre-COVID-19 [13]. Overall, 2,609,911 cases of HAIs were reported each year in the European Union and European Economic Area (EU/EEA) in 2011 to 2012, with an estimated 501 disability-adjusted life years (DALYs) per 100,000 general population [5]. The prevalence of HAIs in the African region ranges from 3% to 15% of patients with considerable under-reporting [14]. Rates for HAIs as low as 1.24% in China [15] compare favourably with those seen in Scotland and among some African countries [13,14]. However, other studies in China have reported higher rates, up to 26.1% of patients in adult intensive care units (ICUs) [16,17]. 

Generally, the burden of HAIs is higher in low- and middle-income countries (LMICs) compared with high-income countries. This arises because of typically suboptimal adherence to infection prevention and control guidelines, lack of effective policies to prevent HAIs, scarcity of strict regulations and their monitoring, inadequate hand hygiene protocols and a lack of antimicrobial stewardship programs (ASPs) [6,18,19,20,21,22]. 

HAIs can develop in a hospital setting in patients receiving health care after 48 h or more after hospital admission [23,24]. Invasive devices, which include vascular devices, urinary catheters and ventilators employed in the medical care of patients, are associated with these HAIs [25,26,27]. The risk of HAI transmission is increased with the lack of awareness and training among health care providers especially in LMICs [28,29,30,31,32]. According to the Center for Diseases Control and Prevention (CDC), common HAIs include ventilator-associated pneumonia, central line-associated bloodstream infections, catheter-associated urinary tract infections and surgical site infections (SSIs) [33].

The COVID-19 pandemic considerably impacted on healthcare delivery systems, appreciably increasing hospitalizations, especially in ICUs [34,35,36,37]. The first case of COVID-19 was identified in Wuhan, China, in December 2019, and spread rapidly after that [38]. However, its impact on morbidity and mortality varied across countries depending on the rapidity and the extent with which lockdown and quarantining measures were introduced [39,40,41,42]. In any event, there was a prolongation of hospital stay associated with COVID-19 in the early stages of the pandemic with no effective vaccines or medicines available to reduce the severity of COVID-19 [42,43,44]. This increased the prevalence of HAIs especially amongst patients in critical care, including those in LMICs [45]. Moreover, the respiratory failure associated with severe or critical COVID-19 is most prevalent in those who are already compromised by age, obesity or comorbidities (including chronic lung disease, diabetes and immunosuppression) and leads to the use of immunosuppressive and immunomodulatory therapies along with mechanical ventilation, all of which further increase the risk of HAIs [4,45,46,47].

Pakistan is a low-middle income South Asian country where the COVID-19 pandemic has appreciably impacted morbidity and mortality, with five different disease waves reported since the first reported case on 26 February 2020 [48,49]. A previous study from Pakistan reported that most of the cases of patients with COVID-19 were mild to moderate; however, older patients with underlying chronic illness may develop severe disease leading to hospitalization as well as mechanical ventilation in intensive care units [50]. By the end of September 2022 over 30,000 deaths due to COVID-19 had been reported in Pakistan [51].

To date, there appears to be no study conducted in Pakistan to ascertain the prevalence of HAIs among COVID-19 patients admitted to ICUs. This is important given the increased morbidity and mortality among such patients [4,25,52]. Consequently, we sought to address this by conducting a retrospective, multicentre study among patients with COVID-19 admitted to the ICUs of hospitals designated for COVID-19 care in the province of Punjab, Pakistan. Punjab Province was chosen for this study as it currently accounts for more than half of the population of Pakistan [53]. In addition, previous research conducted in this province has shown a high consumption of antibiotics among patients with actual or suspected COVID-19 in successive waves [48,53]. This is a concern if such prescribing is inappropriate as this will increase antimicrobial resistance (AMR), with its subsequent impact on morbidity, mortality and costs [54,55,56,57]. There are already concerns with rising AMR rates in Pakistan, with ongoing initiatives to address this through the national action plan (NAP) [58,59]. However, there are ongoing challenges with implementing the NAP in Pakistan, which need to be addressed going forward [60]. 

## 2. Results

A total of 8740 suspected (results waiting) and/or confirmed (by RT-PCR) COVID-19 patients visited or were brought to the designated hospitals during the study period, with 4534 COVID-19 positive patients subsequently hospitalized. Of these, 678 patients were admitted to the ICUs. A total of 636 cases were subsequently included in the final analysis (Figure 1).

The demographic details and clinical features of the COVID-19 patients admitted to the ICUs with or without HAIs are shown in Table 1. The majority of the study population were male (62.6%), aged above 50 years (76.3%) and had multiple symptoms including a fever, cough, sore throat and body ache. Two thirds (69.3%) of patients had severe COVID-19 while 23.9% were critical. Abnormal chest X-ray findings were present in 92.9% of the patients in the ICUs, whereas white blood cells and C-reactive protein elevations were seen in 78.1% and 72.6% of patients, respectively. Additionally, abnormalities in serum ferritin and D-dimer were present in 50.3% and 67.9% of patients, respectively. The development of HAIs was more common in those >50 years (*p* = 0.039) and patients who presented with critical diseases. 

A total of 67 HAIs from different microorganisms developed among the 62 patients with HAIs on the ICU giving a prevalence of 9.7%. Out of the 67 HAIs, 23 were ventilator-associated lower respiratory tract infections, 16 catheter-related urinary tract infections, 14 catheter-related blood stream infections, 12 blood infections of unknown origin and two were fungal infections (Figure 2).

Details of the pathogens causing these HAIs are presented in Figure 3. *Staphylococcus aureus*, *pseudomonas aeruginosa* and *Klebsiella pneumoniae* were identified in 18, 17 and 13 patients, respectively.

The use of antipyretics, corticosteroids and antibiotics was universal (100%) in the studied cases (Table 1). Nearly 80% of the patients received anticoagulants, whereas 18.2% received antithrombotic agents. The use of remdesivir and tocilizumab was seen in 57.2% and 11.5% of patients, respectively. A significantly higher number of patients who developed HAIs were on anticoagulants (*p* = 0.003), antithrombotic agents (*p* < 0.001), antivirals (*p* < 0.001) and IL-6 inhibiting agents (*p* < 0.001) compared with the non-HAI group. In the case of anticoagulants, it was not possible to differentiate their prescribing for either prophylaxis or treatment. The rate of secondary infection was significantly higher in patients who were on invasive mechanical ventilation (*p* < 0.001), had central venous access (*p* = 0.023), or had a urinary catheter placed (*p* < 0.001).

The predictors of HAI development are shown in Table 2. In our multivariable regression model, the prescribing of tocilizumab, and the presence of urinary catheters were found to be independent predictors of HAIs (Table 2).

As shown in Table 3, patients who developed a secondary infection had a significantly longer length of stay in the ICU (*p* < 0.001). Overall, 96% of patients admitted to the ICU with COVID-19 had a complete recovery and were discharged from the hospital, whereas 3.6% (*n* = 23) died. The mortality rate though was significantly higher in patients who developed HAIs (25.8% vs. 1.2%, *p* < 0.001).

In our multivariable-adjusted model using covariates that were significantly associated with mortality in the univariate analysis, e.g., COVID-19 severity (moderate-severe vs. critical), underlying heart disease, invasive mechanical ventilation, presence of a urinary catheter, and HAI development, HAIs (OR 11.18, 95% CI 3.65–34.28, *p* < 0.001) and heart disease (OR 3.67, 95% CI 1.23–10.99, *p* = 0.020) were the independent predictors of mortality.

## 3. Discussion

We believe this is one of the first studies conducted among COVID-19 patients admitted to ICUs in Pakistan to evaluate the predictors and outcomes of HAIs. The prevalence of HAIs among patients with COVID-19 admitted to the ICUs in our study was 9.7%. This compares with the findings of He et al. (2020) just after the emergence of the COVID-19 pandemic, who indicated the prevalence of HAIs among all patients admitted with COVID-19 was 7.1% in a single centre in Wuhan, China [61]. In a study reported from Singapore, the incidence of HAIs among hospitalized patients with COVID-19 treated in ICUs was 14.3% [62], with similarly low rates (6.1%) seen in the UK for bacterial co-infections among patients in ICU with COVID-19 [63]. This contrasts with the study of Grasselli et al. (2021), who reported appreciably higher prevalence rates of HAIs among COVID-19 patients admitted to the ICUs among participating Italian hospitals at 46% [4]. Similarly, Falcone et al. (2020) reported high rates at 71.6% among patients with COVID-19 admitted to ICUs in participating Italian hospitals [64]. Our findings also contrast with those of Ghali et al. (2021) in Tunisia where HAIs were seen in 35.1% of patients in ICU with COVID-19 [24], and with de Hesselle et al. (2022) among patients critically ill in ICUs enrolled into the Lean European Open Survey on SARS-CoV-2-Infected Patients (LEOSS) study [27]. Secondary bacterial infections were documented in the critical phase in 40.4% of cases in this study, and secondary fungal infections in 14.6% of cases [27]. Other studies though have recorded appreciably higher rates with up to 68% of patients in ICUs acquiring secondary bacterial infections during their stay, principally secondary pneumonia [65]. These observed differences may reflect the indications for admission to ICUs and the severity of admitted patients. Some patients in the ICUs in our study did not require oxygen therapy, and there were also relatively low numbers of patients requiring central venous catheters (Table 1). There has also been improved knowledge regarding managing patients with COVID-19 admitted to hospital during successive waves of the pandemic, including the use of steroids in patients requiring respiratory support [66], which may also influence the findings between studies.

Our study highlighted that the most common HAIs reported among patients with COVID-19 admitted to the ICU were ventilator-associated lower respiratory tract infections, i.e., ventilator-associated pneumonia, followed by catheter-associated urinary tract infections and catheter-related blood stream infections. These findings are similar to those of Grasselli et al. (2021) [4], with high rates of HAIs seen among ventilated patients in the studies of de Hesselle et al. (2022) [27], Bardi et al. (2021) [52] and Falcone (2020) et al. [64]. There findings are also similar to a study from Tunisia where frequent HAIs reported among hospitalized COVID-19 patients admitted to ICU were pneumonia/ lower respiratory tract infections followed by urinary tract infections [24]. Following guidelines, including following recommendations regarding catheter placements and care, can help to reduce these HAIs [3]. This reflects increasing recognition of adherence to standard guidelines among key stakeholder groups as a measure of the quality of care provided [67,68,69,70].

The most frequently isolated microorganisms associated with HAIs in our study were *Staphylococcus aureus*, *Pseudomonas aeruginosa* and *Klebsiella pneumoniae*, which is similar to other studies; however, others have reported differences [71,72,73]. Our findings are also similar to those of a previous study from the USA [74], with a study from Italy highlighting that *Acinetobacter baumannii* were most frequent cause of HAIs among critically ill hospitalized COVID-19 patients [4]. Differences in reported studies among patients in ICUs may reflect differences in endemic bacteria and local antibiotic prescribing practices as well the case mix differences between the various studies, with more mild and moderate cases in some studies along with a smaller number of ventilated patients.

Our study also revealed that HAIs among ICU patients developed more often among those aged above 50 years and with comorbidities including diabetes mellitus and hypertension, as well as with COVID-19 severity, abnormal chest X-rays and laboratory investigations including C-reactive proteins and white blood cells. This is similar to studies from Belgium, China, Italy and Spain as well as among a number of LMICs [45,52,61,64,65]. Others risk factors involved with HAI development among COVID-19 patients admitted to ICU in our study were those prescribed anticoagulants, antithrombotic, antivirals and IL-6 inhibitors including tocilizumab, with tocilizumab typically prescribed to severe or critical patients. In these circumstances though, it is not possible to attribute the greater risk of HAI to the recognised immunosuppressant effects of tocilizumab. In contrast to the findings from our study, a study from the USA found that key risk factors for HAIs for patients admitted to hospital with COVID-19 were dexamethasone and antibiotic exposure at the time of hospital admission [75]. However, this was for all patients and not just those admitted to ICUs. Ascertaining the effect of both steroids and prior or current antibiotic use on HAI risk was not possible in our study due to the universal use of both in the ICU population.

Similar to other studies [4,45,52], we observed higher mortality amongst COVID-19 patients who developed HAIs compared with those who did not develop HAIs. However, the influence of other factors such as disease severity and need for mechanical ventilation were not controlled.

Finally, we are aware that 100% of COVID-19 patients in the ICU were prescribed antibiotics irrespective of whether they had an HAI or not. This likely reflects clinical uncertainty and concern for bacterial co-infection in patients who are severely unwell and vulnerable to acute deterioration. High antibiotic prescribing among patients admitted to hospital with COVID-19 has been observed widely in Pakistan and across countries and is in direct contrast with the low rates of bacterial co-infection observed in numerous studies [48,53,76,77,78]. High rates of unnecessary antimicrobial prescribing risk enhancing AMR and contribute further to HAI risk [48,53,54,57,75,76]. This is of particular concern in Pakistan where there are already high rates of AMR [53,54,59], and should be the focus of further studies alongside antimicrobial stewardship interventions. We will be following this up in future studies.

We are aware that there are a number of limitations with our study. Firstly, we collected data from only three public sector tertiary care hospitals. Consequently, we cannot generalize our findings to the whole of Pakistan. Secondly, we did not collect information from private sector hospitals. However, this was deliberate as patients admitted to ICUs in tertiary public hospitals are likely to be more critically ill than those in private or lower-level public hospitals. Despite these limitations, we believe our findings will facilitate clinicians, public health experts and policy makers regarding potential measures to reduce HAIs among hospitalized patients particularly those with COVID-19. This information can subsequently lead to the development of possible quality improvement programmes as part of ASPs among ICUs in Pakistan to reduce future HAI rates.

## 4. Materials and Methods

### 4.1. Study Design and Setting

A retrospective review of medical records was conducted among patients admitted to the ICU due to COVID-19, in three purposely selected public sector tertiary care hospitals, designated for COVID-19 in the Punjab province. Data was collected for all COVID-19 ICU patients over a five-month period from April to August 2021, which was during the third and fourth COVID-19 waves in Pakistan. This methodology involving a retrospective review of patient records is similar to other studies involving the co-authors [48,53,79,80,81,82,83].

All three hospitals were equipped with the necessary equipment and devices, including mechanical ventilators, laboratory facilities and medicines to treat patients with COVID-19.

### 4.2. The Data Collection Form

A data collection form was designed based on published studies, as well as current guidelines [21,23,26,84,85], to collect data of ICU patients admitted due to COVID-19. The following variables were included in the data collection form: (1) Demographic characteristics: age of the patient in years, their gender and residence; (2) Total number of days in ICU, presence of any comorbidities, clinical signs and symptoms, and oxygen use. The presence of co-morbidities is important since we were aware that some patients were admitted to ICU without the need for oxygen therapy; however they had comorbidities including pre-existing respiratory illness, were immunocompromised, or had underlying cardiac diseases or diabetes mellitus, whilst potentially waiting for laboratory results; (3) COVID-19 severity: Assessed as asymptomatic, mild, moderate, severe or critical as per the guidelines issued by the Ministry of Health or as per guidelines issued by National Health Services, Regulation and Coordination, Government of Pakistan [84]. Those patients who had oxygen saturation below 94% but above 90%, and chest X-rays with infiltrates involving <50% of the total lung fields, were declared as moderate cases. Severe cases were those who had a fever and cough along with respiratory rate < 30, severe respiratory distress, chest X-ray with infiltrates involving <50% of the total lung fields, and oxygen saturation ≤ 90 on room air. The presence of acute respiratory distress syndrome (ARDS) or worsening of respiratory symptoms, bilateral opacities or lung collapse in chest X-rays or CT scans and respiratory or cardiac failure were considered as critical cases. Moderate, severe and critical COVID-19 patients were admitted to the ICU depending upon other disease manifestations; (4) Laboratory findings including chest X-rays, white blood cell (WBCs) counts, C-reactive protein (CRP) levels, D-dimer and serum ferritin levels were also documented. The X-ray findings were reviewed by medical doctors and the treating physicians were consulted in case of any confusion. Normal ranges/presence or absence of WBCs and CRP, D-dimer and serum ferritin were taken from the references mentioned on the testing kits; (5) Medicines prescribed at the time of admission to the ICU were also documented including antipyretics, antihistamines, anticoagulants, antithrombotic medicines and antibiotics; (6) Devices used in the ICU to treat patients, including any invasive mechanical ventilation, central venous catheters, urinary catheters and any orotracheal tubes; (7) Presence of HAIs. HAIs were defined as an infection appearing ≥48 h after hospital admission unless patients had been discharged from the hospital [86,87]. The type or incidence of HAIs were confirmed using the methodology of the European Centers for Diseases Control and Prevention (ECDC) [5,85,88,89]. If HAIs were present, the type of HAI along with the microorganism associated with the HAI [85,89,90]; (8) Outcomes: Whether the patient was discharged from hospital or died.

The draft data collection form was tested in a pilot study involving 30 patients to see if the developed forms were able to capture the necessary data sets for the study. These patients were subsequently excluded from the full study. Based on the pilot study results, no modification to the data collection form was necessary.

### 4.3. The Data Collection Procedure

The team of investigators (including medical doctors and pharmacists) visited the three hospitals after permission was granted by the hospital administration. They were provided with full training by the principal investigator (ZUM) before commencing the data collection process. The medical records of patients were accessed and reviewed thoroughly to obtain the required information. All the records were in a paper-based format and in case of any uncertainties, healthcare professions were requested for subsequent interpretation.

### 4.4. Inclusion and Exclusion Criteria

The inclusion criteria were any patient with confirmed COVID-19 (confirmed via RT-PCR testing) admitted to the ICU between April and August 2021, among the three selected hospitals, who had a complete medical record available in the record room. All other patients admitted to the hospitals with COVID-19, but who were not admitted to the ICU, those admitted to the ICU with COVID-19 but outside the time period for the study, and those with incomplete medical records, were excluded from this study.

### 4.5. Data Analysis

The data were entered onto a Microsoft Excel^®^ sheet, cleaned and after coding imported into the Statistical Package for the Social Sciences (SPSS Inc., version 22, IBM, Chicago, IL, USA). The mean and standard deviation were calculated for continuous variables whereas frequency/number (N) and percentage (%) were calculated for categorical variables. Demographic data, clinical features of COVID-19 patients, and outcomes were compared between patients with HAIs and those with no HAI using χ^2^ test and Fischer’s exact test as appropriate. We performed binary logistic regression to determine the factors associated with the development of HAIs among COVID-19 patients as well as for the predictors of mortality. A *p* < 0.05 (two-sided) was considered statistically significant.

### 4.6. Ethical Considerations

Ethical approval was obtained from the Office of Research, Innovation and Commercialization (ORIC), Lahore College for Women University (LCWU), Lahore, Pakistan (Ref. no. ORIC/LCWU/395). Ethical approval of this study was also obtained from the ethics committees of the participating hospitals.

Since all data was obtained from medical records and drug prescription charts without patient contact, written consent was not required. This is similar to other retrospective and point prevalence surveys conducted by the co-authors [53,80,91,92,93,94,95]. Furthermore, no personal data was recorded, and any patient data recorded was de-identified. Participants’ data were subsequently coded and stored in a password-protected Microsoft Excel^®^ sheet accessible only to the researchers in order to ensure patient confidentiality.

## 5. Conclusions and Recommendations

There is a high prevalence of HAIs among COVID-19 patients admitted to ICUs in Pakistan. This is associated with higher mortality, although mortality rates are lower than seen in a number of other countries potentially reflecting lower rates of mechanical ventilation and central venous catheters in the cohort studied. Independent factors associated with an increase in HAIs were the use of urinary catheters and mechanical ventilation. Tocilizumab was also independently associated with higher risk of HAI; however, this may reflect disease severity and longer hospital stay and requires further study. By adopting core elements of infection prevention and control, the prevalence as well as the mortality associated with HAIs can be minimized. This includes reviewing key issues and approaches surrounding the use of catheters alongside mechanical ventilation. Moreover, a multisectoral approach under a targeted antimicrobial stewardship programme (ASP) can help address inappropriate antimicrobial use among critically ill patients admitted to the ICU. We will now be working with key personnel in these hospitals to seek ways to reduce HAIs among ICUs in Pakistan through instigating appropriate ASPs and will be following this up in future studies.

## Figures and Tables

**Figure 1 antibiotics-11-01806-f001:**
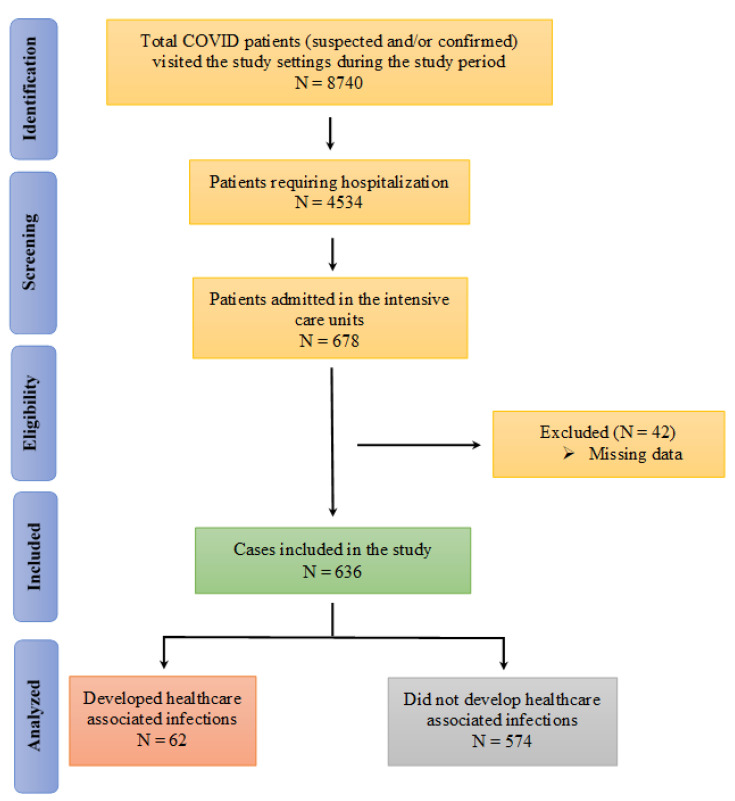
Flow diagram of the identification of the patients for inclusion in the study.

**Figure 2 antibiotics-11-01806-f002:**
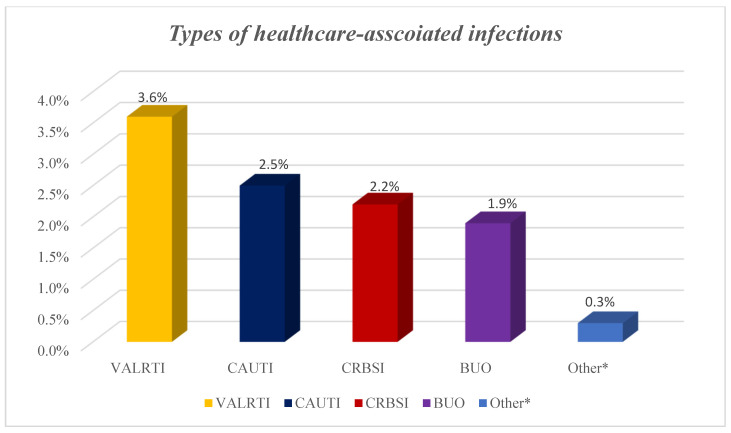
Types of healthcare-associated infections among COVID-19 patients. NB: BUO: bloodstream infections of unknown origin; CAUTI: catheter-associated urinary tract infection; CRBSI: catheter-related blood stream infection; HAI: healthcare-associated infections; VALRTI: ventilator-associated lower respiratory tract infections; Others* included fungal infections.

**Figure 3 antibiotics-11-01806-f003:**
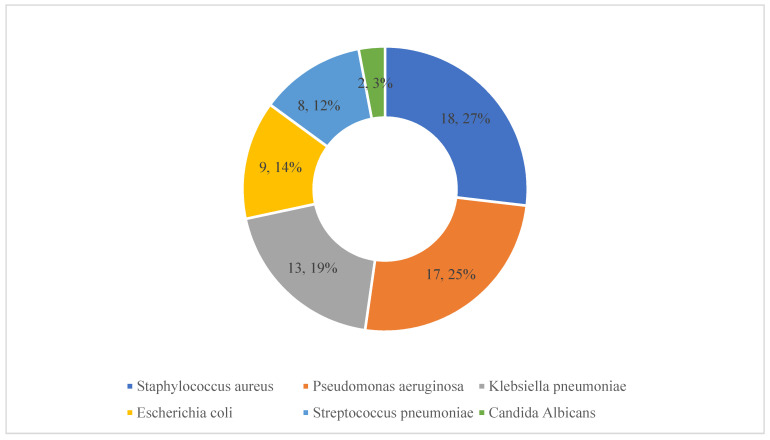
Details of microorganisms responsible for healthcare-associated infections. Note: The numbers refer to the number of patients with these isolates along with the percentages (*n* = 67).

**Table 1 antibiotics-11-01806-t001:** Demographics and clinical features of COVID-19 patients with and without healthcare-associated infections (HAIs).

Variable	N (%)			*p*-Value
	Overall	HAI (N = 62)	No HAI (N = 574)	
**Age (years)**				0.039
<30	21 (31.3)	0 (0.0)	21 (3.7)	
31–50	130 (20.4)	7 (11.3)	123 (21.5)	
>50	485 (76.3)	55 (88.7)	430 (74.9)	
**Sex**				0.784 *
Male	398 (62.6)	40 (64.5)	358 (62.4)	
Female	238 (37.4)	22 (35.5)	216 (37.6)	
**Residence**				0.673 *
Rural	423 (66.5)	43 (69.4)	380 (66.2)	
Urban	213 (33.5)	19 (30.6)	164 (33.8)	
**Comorbidities**				
Diabetes	142 (22.3)	16 (25.8)	126 (22.0)	0.521 *
Hypertension	127 (20.0)	13 (21.0)	114 (19.9)	0.867 *
Heart disease	64 (10.1)	11 (17.7)	53 (9.2)	0.044 *
Respiratory disorder	38 (6.0)	4 (6.5)	34 (5.9)	0.780 *
Other ^†^	10 (1.6)	1 (1.6)	9 (1.6)	1.000 *
**Symptoms at presentation**				0.144
Fever + cough	24 (3.8)	4 (6.5)	20 (3.5)	
Fever + myalgia	65 (10.2)	5 (8.1)	60 (10.5)	
Fever + sore throat	100 (15.7)	5 (8.1)	95 (16.6)	
Fever + dyspnea	122 (19.2)	7 (11.3)	115 (20.0)	
Fever + cough + dyspnea	97 (15.3)	13 (21.0)	84 (14.6)	
Fever + sore throat + body ache	59 (9.3)	7 (11.3)	52 (9.1)	
More than three symptoms	169 (26.6)	21 (33.9)	148 (25.8)	
**COVID severity ^††^**				<0.001
Moderate	43 (6.8)	1 (1.6)	42 (7.3)	
Severe	441 (69.3)	12 (19.4)	429 (74.7)	
Critical	152 (23.9)	49 (79.0)	103 (17.9)	
**Laboratory findings ^††^**				
Abnormal X-ray	591 (92.9)	61 (98.4)	530 (92.3)	0.112 *
Abnormal WBC	497 (78.1)	56 (90.3)	441 (76.8)	0.014 *
Abnormal CRP	462 (72.6)	56 (90.3)	406 (70.7)	0.001 *
Abnormal D-Dimer	432 (67.9)	51 (82.3)	381 (66.4)	0.010 *
Abnormal Serum ferritin	320 (50.3)	46 (74.2)	274 (47.7)	<0.001 *
**Oxygen therapy**				0.150*
Yes	582 (91.5)	60 (96.8)	522 (90.9)	
No ^††^	54 (8.5)	2 (3.2)	52 (9.1)	
**Medications**				
Antipyretic **	636 (100.0)	62 (100.0)	574 (100.0)	--
Antihistamine	476 (74.8)	51 (82.3)	425 (74.0)	0.169 *
Anticoagulant	506 (79.6)	58 (93.5)	448 (78.0)	0.003 *
Antithrombotic	92 (14.5)	30 (48.4)	62 (10.8)	<0.001 *
Antitussive	377 (59.3)	39 (62.9)	338 (58.9)	0.588 *
Corticosteroid **	636 (100.0)	62 (100.0)	574 (100.0)	--
Antibiotic **	636 (100.0)	62 (100.0)	574 (100.0)	--
Antiviral (remdesivir)	364 (57.2)	52 (83.9)	312 (54.4)	<0.001 *
IL-6 inhibitor (tocilizumab)	73 (11.5)	50 (80.6)	23 (4.0)	<0.001 *
Vitamin	362 (56.9)	37 (59.7)	325 (56.6)	0.687 *
**Devices used in the ICU**				
Invasive mechanical ventilation ^†††^	111 (17.5)	37 (59.7)	74 (12.9)	<0.001 *
Central venous catheter	7 (1.1)	3 (4.8)	4 (0.7)	0.023 *
Urinary catheter	157 (24.7)	28 (45.2)	129 (22.5)	<0.001 *
Orotracheal tube	117 (18.4)	17 (27.4)	100 (17.4)	0.59 *

NB: ^†^ Other diseases included gastroesophageal reflux disease, peptic ulcer, constipation, arthritis, and urticarial; ^††^ See Methods for definitions and rationale; ^†††^ no data on CPAP or HFNO; * Fisher’s exact test; ** No statistics were computed because the variable was a constant.

**Table 2 antibiotics-11-01806-t002:** Factors associated with the development of healthcare-associated infections (HAIs) in COVID-19 patients.

Covariates	B	SE	Wald	df	*p*-Value	Odds Ratio	95% Confidence Interval	
							Lower Bound	Upper Bound
Age *	−0.589	0.623	0.894	1	0.345	0.555	0.163	1.882
Heart disease	−1.031	0.629	2.686	1	0.101	0.357	0.104	1.224
**COVID Severity ^†^**								
Moderate (Reference)								
Severe	−0.423	1.171	0.131	1	0.718	0.655	0.066	6.503
Critical	1.772	1.290	1.887	1	0.169	5.884	0.469	73.748
Anticoagulant use	−0.303	0.813	0.139	1	0.710	0.739	0.150	3.633
Antithrombotic agents	−0.332	0.562	0.350	1	0.554	0.717	0.238	2.158
Antiviral	0.002	0.694	0.000	1	0.998	1.002	0.257	3.905
Tocilizumab ^††^	4.437	0.482	84.920	1	<0.001	84.559	32.906	217.291
Invasive mechanical ventilation	0.600	0.655	0.839	1	0.360	1.822	0.505	6.576
Central venous line	1.066	1.565	0.464	1	0.496	2.904	0.135	62.450
Urinary catheter	1.124	0.451	6.223	1	0.013	3.077	1.272	7.443

NB: B: Unstandardized regression weight; Df: degree of freedom; SE: standard error; * age < 50 years was taken as the reference in the regression model; ^†^ see Methods for definitions; ^††^ typically only prescribed for severe or critical patients.

**Table 3 antibiotics-11-01806-t003:** Outcomes of COVID-19 patients with healthcare-associated infections (HAIs) in ICUs.

Outcome	N (%)			*p*-Value
	Total	HAI (N = 62)	No HAI (N = 574)	
**Length of ICU stay (days)**				<0.001
≤7	44 (6.9)	1 (1.6)	43 (7.5)	
8–14	69 (10.8)	3 (4.8)	66 (11.5)	
15–21	226 (35.5)	9 (14.5)	217 (37.8)	
22–29	159 (25.0)	13 (21.0)	146 (25.4)	
≥30	138 (21.7)	36 (58.1)	102 (17.8)	
**Outcome ^†^**				<0.001 *
Discharged alive	613 (96.4)	46 (74.2)	587 (98.8)	
Deceased	23 (3.6)	16 (25.8)	7 (1.2)	

NB: * Fisher’s exact test; ^†^ No comparison with outcomes of patients with hospital-acquired infections (HAIs) in other wards in the surveyed hospitals.

## Data Availability

Further data is available from the co-authors on reasonable request.

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
