# Peer review of "Predictors and Outcomes of Healthcare-Associated Infections among Patients with COVID-19 Admitted to Intensive Care Units in Punjab, Pakistan; Findings and Implications"

_antibiotics, 2022, doi:10.3390/antibiotics11121806_

Round 1

Reviewer 1 Report

This research  review the HAIs for hospitalized COVID-19 patients. The research methods and the results are adequate and can be used as an reference for care COVID-19 patients.

I suggest to accept this paper without further suggestions.

Author Response

Comments and Suggestions for Authors

This research review the HAIs for hospitalized COVID-19 patients. The research methods and the results are adequate and can be used as an reference for care COVID-19 patients.

I suggest to accept this paper without further suggestions.

Author comments: Thank you for your kind comments – very much appreciated! 

Reviewer 2 Report

1.    The period of time for the study is short, and as we are almost at the end of 2022 maybe the author could consider adding more data to increase the impact of the study.

2.    The article stops short of comparing the outcome to patients without COVID -19 infection at the ICU.

3.    Please revise figure 1 legend, there is abbreviation mentioned that are not in the figure

4.    In Table 1 also, please remove (**) because there is none in the table. In the same table you can highlight significant p value in bold to make it easy for the reader to identify.

5.    In figure 3 it is helpful to include the % of patients with the isolates as well.

6.    Please do not repeat words, like: Finally, in the discussion

Author Response

Comments and Suggestions for Authors 

  1. The period of time for the study is short, and as we are almost at the end of 2022 maybe the author could consider adding more data to increase the impact of the study.

Author comment: Thank you. This was the first study of its kind in Pakistan with COVID-19 patients. We are now planning a follow-up study given some of the areas of concern – building on these initial results. We hope this is now acceptable.

  1. The article stops short of comparing the outcome to patients without COVID -19 infection at the ICU.

Thank you. As you saw, the current study was conducted among patients with COVID-19 admitted to ICU which was the main focus. Consequently, we didn’t collect any information regarding patients admitted to ICU due to other conditions. In our next studies, we will look to address this. We hope this is acceptable to you.

  1. Please revise figure 1 legend, there is abbreviation mentioned that are not in the figure.

Author comments: Thank you, the suggested revision has been made in the figure.

  1. In Table 1 also, please remove (**) because there is none in the table. In the same table you can highlight significant p value in bold to make it easy for the reader to identify.

Author comment: Thank you. We beg to differ as the ** are by antipyrectics, etc. under medications – explained in the NB. We hope this is OK.

  1. In figure 3 it is helpful to include the % of patients with the isolates as well.

Author comments: Thank you, the suggested revision has been made in the Figure 3.

  1. Please do not repeat words, like: Finally, in the discussion

Author comments: Thank you – now altered. We have also been through the manuscript with the help of English-speaking co-authors – one of whom has published over 450 papers in peer-reviewed Journals. We hope this is now OK.

Reviewer 3 Report

The manuscript entitled as “Predictors and outcomes of healthcare associated infections among patients with COVID-19 admitted to intensive care units in Punjab, Pakistan; findings and implications” by Zia Ul Mustafa et al is very interesting  due to its clinical importance for the clinical practitioners working on the treatment of the COVID patients. In fact, the current proposal is interesting.  Therefore, I recommend the publication of the present study after some minor revisions. Some suggestions that will fatherly improve the manuscript:

1.       Add one line concise conclusion statement in the abstract section.

2.       Add a few lines on the history of COVID-19 in the introduction section.

3.       Add future recommendations of the study.

4.       Recheck the spellings and remove the grammatical mistakes.

Author Response

Comments and Suggestions for Authors

The manuscript entitled as “Predictors and outcomes of healthcare associated infections among patients with COVID-19 admitted to intensive care units in Punjab, Pakistan; findings and implications” by Zia Ul Mustafa et al is very interesting due to its clinical importance for the clinical practitioners working on the treatment of the COVID patients. In fact, the current proposal is interesting.  Therefore, I recommend the publication of the present study after some minor revisions. 

Author comments: Thank you for these comments – very much appreciated!

Some suggestions that will fatherly improve the manuscript:

  1. Add one-line concise conclusion statement in the abstract section

Author comment: Thank you. We have added this in updated version of manuscript.

  1. Add a few lines on the history of COVID-19 in the introduction section.

Author comments: Thank you – this has now been added. We hope this is now OK.

  1. Add future recommendations of the study.

Author comments: Thank you. We have added this in updated version of manuscript.

  1. Recheck the spellings and remove the grammatical mistakes.

Author comments: Thank you. We have also been through the manuscript with the help of English-speaking co-authors – one of whom has published over 450 papers in peer-reviewed Journals. We hope this is now OK.

Reviewer 4 Report

Dear authors,
the topic of the manuscript is interesting, but some changes are needed. 

INTRODUCTION
- Please provide a synonym of delivery at Line 50.

- Please provide uniformity in numbers editing between lines 54 and 60.
- Please clarify the meaning of the sentence with ref. [15] at Line 66.

- Please modify sentence at Lines 69-72. (verb missing?)

MATERIALS and METHODS

-Please evaluate to describe in the text the materials and methods section before the results paragraph. 

RESULTS

- Please clarify the meaning of "suspected" and "confirmed" at Line 115.
-Please change the "NB:2 beneath figures and tables with "Notes:".
-Please clarify English language in sentences at Lines 206-209 and 213-214. 
-Please if available include extensive data on developed AMR (should be interesting for the readers)

DISCUSSION

- Please change "Finally" at Line 260 (There is a Finally in Line 266 too).

CONCLUSIONS

_please provide a clear description of the new strategies cited in Lines 391-392.

Furthermore, extensive English language improvement is needed. 

Author Response

Dear authors, the topic of the manuscript is interesting, but some changes are needed.

Author comments. Thank you for your help. We hope we have adequately addressed these.

  1. INTRODUCTION

  1. Please provide a synonym of delivery at Line 50.

Author comments: Now amended.

  1. Please provide uniformity in numbers editing between lines 54 and 60.

Author comments: Thank you – now updated

  1. Please clarify the meaning of the sentence with ref. [15] at Line 66.

Author comments: Thank you now clarified

  1. Please modify sentence at Lines 69-72. (Verb missing?)

Comments: Thank you – now amended.

  1. MATERIALS and METHODS -Please evaluate to describe in the text the materials and methods section before the results paragraph.

Author comments: Thank you for this. However, as you are aware – the Template for submission of papers to Antibiotics places Methods after Results and Discussion. I trust this is acceptable to you.

  1. RESULTS

  1. a) Please clarify the meaning of "suspected" and "confirmed" at Line 115.

Author comments: Thank you. We have added this in updated version of manuscript.

  1. b) Please change the "NB:2 beneath figures and tables with "Notes:"

Author comments: Thank you – now done.

  1. c) Please clarify English language in sentences at Lines 206-209 and 213-214.

Author comments: Thank you – now amended. We have also been through the manuscript with the help of English-speaking co-authors – one of whom has published over 450 papers in peer-reviewed Journals. We hope this is OK.

  1. d) Please if available include extensive data on developed AMR (should be interesting for the readers)

Author comments: Thank you. We did not collect this data during the current project; consequently, have removed one paragraph to reduce confusion. We will though collect this data in our next projects to help give further guidance, and hope this is now acceptable.

  1. DISCUSSION - Please change "Finally" at Line 260 (There is a Finally in Line 266 too).

Author comments: Thank you – now amended.

  1. CONCLUSIONS _please provide a clear description of the new strategies cited in Lines 391-392.

Author comments: Thank you now addressed. We hope this is now OK.

  1. Furthermore, extensive English language improvement is needed.

Author comments: Thank you – we have now been through the paper with the help of two native English speakers – one of whom has over 450 publications in peer-reviewed Journals. We hope this is now OK

Round 2

Reviewer 4 Report

The manuscript can be accepted in the present form. Consider to place materials and methods section as second paragraph.